# 5-Aza-2′-Deoxycytidine and Valproic Acid in Combination with CHIR99021 and A83-01 Induce Pluripotency Genes Expression in Human Adult Somatic Cells

**DOI:** 10.3390/molecules26071909

**Published:** 2021-03-29

**Authors:** Alain Aguirre-Vázquez, Luis A. Salazar-Olivo, Xóchitl Flores-Ponce, Ana L. Arriaga-Guerrero, Dariela Garza-Rodríguez, María E. Camacho-Moll, Iván Velasco, Fabiola Castorena-Torres, Nidheesh Dadheech, Mario Bermúdez de León

**Affiliations:** 1Centro de Investigación Biomédica del Noreste, Departamento de Biología Molecular, Instituto Mexicano del Seguro Social, Monterrey, Nuevo León 64720, Mexico; alain.aguirre@ipicyt.edu.mx (A.A.-V.); anarriaga@live.com (A.L.A.-G.); dmgr.040794@gmail.com (D.G.-R.); maria.camachom@udem.edu (M.E.C.-M.); 2Departamento de Biología Molecular, Instituto Potosino de Investigación Científica y Tecnológica, San Luis Potosí, San Luis Potosí 78216, Mexico; olivo@ipicyt.edu.mx; 3Instituto de Fisiología Celular-Neurociencias, Universidad Nacional Autónoma de México, Ciudad de México 04510, Mexico; xochitlflores91@gmail.com (X.F.-P.); ivelasco@ifc.unam.mx (I.V.); 4Laboratorio de Reprogramación Celular, Instituto Nacional de Neurología y Neurocirugía “Manuel Velasco Suárez”, Ciudad de México14269, Mexico; 5Facultad de Ciencias Biológicas, Universidad Autónoma de Nuevo León, UANL, San Nicolás de los Garza, Nuevo León 66451, Mexico; 6Escuela de Medicina, Tecnologico de Monterrey, Monterrey, Nuevo León 64710, Mexico; fcastorena@tec.mx; 7Clinical Islet Transplant Program, Alberta Diabetes Institute, University of Alberta, Edmonton, AB T6G2E1, Canada; dadheech@ualberta.ca

**Keywords:** reprogramming, 5-aza-2′-deoxycytidine, valproic acid, stem cells, epigenetics

## Abstract

A generation of induced pluripotent stem cells (iPSC) by ectopic expression of OCT4, SOX2, KLF4, and c-MYC has established promising opportunities for stem cell research, drug discovery, and disease modeling. While this forced genetic expression represents an advantage, there will always be an issue with genomic instability and transient pluripotency genes reactivation that might preclude their clinical application. During the reprogramming process, a somatic cell must undergo several epigenetic modifications to induce groups of genes capable of reactivating the endogenous pluripotency core. Here, looking to increase the reprograming efficiency in somatic cells, we evaluated the effect of epigenetic molecules 5-aza-2′-deoxycytidine (5AZ) and valproic acid (VPA) and two small molecules reported as reprogramming enhancers, CHIR99021 and A83-01, on the expression of pluripotency genes and the methylation profile of the OCT4 promoter in a human dermal fibroblasts cell strain. The addition of this cocktail to culture medium increased the expression of OCT4, SOX2, and KLF4 expression by 2.1-fold, 8.5-fold, and 2-fold, respectively, with respect to controls; concomitantly, a reduction in methylated CpG sites in OCT4 promoter region was observed. The epigenetic cocktail also induced the expression of the metastasis-associated gene S100A4. However, the epigenetic cocktail did not induce the morphological changes characteristic of the reprogramming process. In summary, 5AZ, VPA, CHIR99021, and A83-01 induced the expression of OCT4 and SOX2, two critical genes for iPSC. Future studies will allow us to precise the mechanisms by which these compounds exert their reprogramming effects.

## 1. Introduction

The generation of induced pluripotent stem cells (iPSC) has opened tremendous possibilities for research in regenerative medicine, drug development, basic biology, and cellular transplantation therapy [1]. As originally described by Yamanaka and colleagues, iPSC cells can be reprogramed from somatic cells by the transduction and co-expression of the transcription factors OCT4, SOX2, KLF4, and c-MYC, now known as the Yamanaka factors [2], either by host genome integrating or non-integrating approaches. Among potential challenges to harness the use of iPSC are their low reprogramming efficiency, marked clonal differences, and propensity to off-targeted differentiation due to undetermined variability in pluripotent gene expression.

Despite promising results, there are several concerns still unresolved that limit the potential application of iPSC in human diseases modeling and therapeutic development. These include the dysregulated genomic integration of the Yamanaka factors [3]. For instance, iPSC reprogrammed by genomic integration of Yamanaka factors has the disadvantage of re-expressing the pluripotency genes after a differentiation process to a specific cell-type, which may not be desirable for potential translational medicine applications [4]. Since this groundbreaking research was reported, just over a decade ago, improved methods have been developed to ascertain safer and controlled means of cellular delivery of the Yamanaka factors [5].

Although integrating-free methods promote an encouraging landscape for its potential use in regenerative medicine and subsequently to translational therapies, there are still profound limitations about the use of ectopic expression of these pluripotency genes and a very low reprogramming rate. Recently, the search for clinical-grade iPSC had a significance advance when the use of small molecules replace the necessity of exogenous reprogrammable factors to generate iPSC from mouse embryonic fibroblasts [6]. The advantage of using small molecules that modulate several metabolic pathways, transform energy production of the cell, or modify the epigenome has enhanced the use of high-throughput screening methods to evaluate the effect that several molecules have on the expression of pluripotency genes with the possibility of increasing the reprogramming efficiency.

One of the most critical features during the generation of iPSC is the global remodeling of the cellular epigenome [7]. During the reprogramming process, somatic cells must overcome epigenetic barriers to restructure the genome architecture and reactivate the pluripotency core. Interestingly, changes in epigenome homeostasis are not only responsive to environmental signals, but also commonly-used pharmaceutical drugs [8]. These drugs can modify epigenetic and epigenomic homeostasis by direct or indirect mechanisms [9]. In this perspective, the cytidine analog molecule, 5-aza-2′-deoxycytidine (5AZ), initially used as a chemotherapeutic agent for acute myelogenous leukemia and myelodysplastic syndrome, acts as a demethylating agent [10]. The incorporation of 5AZ during DNA synthesis prevents DNA methylation and promotes epigenetic changes by modifying the histone code [11]. Similarly, valproic acid (VPA), a short-chain fatty acid used to treat epilepsy and bipolar disorder, has been linked to the inhibition of histone deacetylase activity [12]. VPA is a broad inhibitor of class I histone deacetylases activity causing hyperacetylation of histones [13], induces proteasomal degradation of HDAC2 selectively, and contributes to chromatin decondensation and consequently active gene transcription [12]. Interestingly, VPA enhances somatic cell reprogramming, increasing the generation of iPSC [14]. In the present work, looking to increase the reprogramming efficiency, we evaluated the effect of 5AZ and VPA in combination with CHIR99021 and A83-01, two small reprogramming enhancers on the expression of pluripotency genes by human adult dermal fibroblasts.

## 2. Results

### 2.1. Viability Curves in Fibroblasts by Epigenetic Molecules

To determine non-cytotoxic 5AZ and VPA concentrations for human adult dermal fibroblasts (HDF), dose–response curves of these epigenetic molecules were performed at 24, 48, 72, and 96 h. During the entire 96 h period of treatment with DNA methyltransferase inhibitor 5AZ, we observed a significant decrease in viability of 17% at the concentration of 1 µM at 24 h, which did not reduce further with time. Interestingly, despite continuous treatment with 5AZ for an additional period of 72 h, viability was not significantly affected at any concentration (Figure 1A). Next, we observed that concentrations of VPA higher than 2 mM significantly affect the viability of fibroblasts at all tested times. Additionally, even though at 24, 48, and 96 h the concentration range from 0.5 to 2 mM generates cytotoxicity, this was not more than 20% (Figure 1B). Afterward, we evaluate the combined effect of both epigenetic drugs at 1 µM 5AZ and 1 mM VPA (Figure 1C). As expected, we just observed a short reduction of 21% in cell viability. These results show that the selected doses of epigenetic molecules do not cause a drastic reduction in the viability of the human fibroblasts, which indicates that their combined use for subsequential assays will not be limited by effects on cell viability.

### 2.2. Morphological Changes by Epigenetic Molecules

Then we tested whether the combination of epigenetic molecules 5AZ and VPA could cause morphological changes that are characteristic of the reprogramming process. First, as control, a well-established cell culture for reprogramming, human foreskin fibroblasts, were transfected with non-integrating vectors [15]. After 11 days, some cells present signs of mesenchymal to epithelial transition and cell aggregates were observed (Figure 2A). After 16 days, the initial formation of hiPSC-like colonies were noticed (Figure 2B). At day 24, the colonies attained iPSC morphology that is observed by tightly closed and compacted cells attached to each other without intercellular spaces, minimal cytoplasmic content, and prominent nucleolus (Figure 2C). We then expanded and characterized the iPSC colonies, which showed round edges, a similar morphology to those reported [16] for hESCs (Figure 2D–G), and were positive for the pluripotency markers OCT4 (Figure 2H), SOX2 (Figure 2I), and NANOG (Figure 2J). Of note, the episomal vectors do not contain NANOG cDNA, pointing to the initiation of the endogenous pluripotency program by NANOG positivity. After identifying the gradual changes that occur during the reprogramming process, such as increase of size of nucleus, change in nucleus-to-cytoplasm ratio, and formation of colonies with round edges, the next step was to evaluate whether the combination of epigenetic drugs causes morphological changes similar to those observed during the generation of human iPSC, but without the ectopic expression of the Yamanaka factors. To assess this, we tested the small molecules CHIR99021, a GSK3β inhibitor, and A-83-01, a TGFβ inhibitor, along with the two epigenetic drugs. These small molecules have been reported as reprogramming enhancers, and the doses used here were adapted from protocols in previous reports [17,18,19]. We initiated the HDF culture in combination with 1 µM 5AZ, 1 mM VPA, 5 µM CHIR99021, and 0.5 µM A-83-01. We performed a two-stage strategy, based in protocols used to generate chemical induced pluripotent stem cells [6,20], including first treatment with VPA, 5AZ, and the small molecule CHIR99021 containing medium replenished every 48 h. In the second phase, we added A-83-01 to complete the standard medium used to induced and maintain pluripotent stem cells [16] (see Materials and Methods). During the long-term culture of HDF, cells were exposed to the cocktail of epigenetic drugs and small molecules. We did not observe gradual loss of fibroblast-like morphology nor the formation of colonies.

### 2.3. Expression Analysis of Pluripotent Genes by Effect of Epigenetic Molecules

We then asked whether the two-stage strategy, for 30 days, was capable of inducing the expression of pluripotency genes. Quantitative expression analysis of OCT4 and SOX2 genes from treated cells showed a 2.1-fold and 8.5-fold increases in expression, compared to the untreated fibroblasts, although these changes were non-significant. This further revealed that KLF4 gene was significant up-regulated in treated cells with a 2-fold increase. On the other hand, we observed a hypervariable expression of c-MYC and NANOG. Interestingly, a highly significant 9-fold up-regulation of the metastasis-associated gene S100A4 or fibroblast-specific protein 1 was observed after treatment with the cocktail (Figure 3).

### 2.4. Methylation Analysis of OCT4 Promoter by Effect of Epigenetic Molecules

To correlate the upregulation of OCT4 expression by the combination of epigenetic drugs and small molecules with epigenetic changes, we performed a methylation analysis on OCT4 proximal promoter region by bisulfite sequencing. The methylation profile of cytosine-guanine dinucleotides (CpG) in the OCT4 promoter revealed a 44% methylated CpG sites in cells treated with the cocktail of epigenetic drugs and small molecules, compared to 68% methylated CpG sites untreated cells (Figure 4). This result indicates a reduction in DNA methylation levels in the proximal promoter region of OCT4 gene by effect of the cocktail.

## 3. Discussion

We show here that a cocktail containing 5AZ, VPA, CHIR99021, and A83-01 molecules clearly increased the expression of endogenous SOX2, KFL4, and S100A4 genes and non-significantly the expression of OCT4 and NANOG genes in adult human dermal fibroblast cells. Interestingly, it has been reported that adult human and mouse dermal fibroblasts have basal expression levels of some pluripotency genes, and these expression levels can be modified by specific culture conditions [21,22]. Therefore, it is possible that the combination of epigenetic drugs, small molecules, and culture conditions may affect and induce gene expression of OCT4, SOX2, KFL4, and NANOG, although this induction might not be enough to generate morphological iPSC changes. Likewise, there has been some research evidence demonstrating the enhancement in reprogramming efficiency and possible replacement of the Yamanaka factors with epigenetic drugs and small molecules may occur. For instance, 5AZ reset the “epigenetic memory” in mouse iPSC [23], and VPA allows the generation of human iPSC with only two factors, OCT4 and SOX2 [24]. On the other hand, there are reports where the use of trichostatin A, another HDAC inhibitor, contributes to reprogramming efficiency in bovine cloned embryos and Oct4 expression in mouse embryonic stem cells [25,26]. Additionally, the small molecules used in the cocktail, CHIR99021 and A83-01, have been reported to improve the reprogramming efficiency using only the transfection of two factors, OCT4 and KLF4 [18], and as part of one of the first small molecules cocktails used to induce human pluripotent stem cells transduced only with OCT4 [19].

OCT4 is a key transcription factor that has been linked to both pluripotency and oncogenicity [27]. The search for molecules and treatments that can modulate OCT4 expression are of particular importance [25,26]. Interestingly, we observed a decrease in OCT4 promoter methylation status. Shi and collages reported a reduction in methylated CpG sites of the minimal promoter region of OCT4 caused by 5AZ; this compound increased the OCT4 expression in human glioma cells [28]. These findings indicate that 5AZ, which was included in the formulated cocktail, promotes the demethylation of OCT4 promoter region, and prevents the complete silencing of OCT4 in somatic cells. Interestingly, evidence suggests that tumor-initiating cells require OCT4 activation [29,30]. Therefore, additional approaches are needed to elucidate whether this epigenetic modification could impact on the ability of treated cells to form tumors.

As such, NANOG gene expression is associated with the pluripotency network core established by the interactions with OCT4 and SOX2 [31]. Consequently, it is necessary for a pre-iPSC state in which OCT4-SOX2 interaction activates NANOG expression and start the pluripotency network. Unfortunately, in our study, the expression of OCT4 and SOX2 was not enough to significantly reactivate NANOG expression. It has been reported that the activation of the WNT signaling pathway contributes to the induction of pluripotency without the exogenous expression of c-MYC [32]. Regardless, we did not observe any increase in c-MYC expression, despite the stabilization of β-catenin by CHIR99021 [33]. The hypervariable expression of genes can be explained by the fact that these genes are subject to dynamic processes, not static, and only a snapshot of the process itself is observed at the time of evaluation [34].

Surprisingly, we observed a significant 9-fold up-regulation of the metastasis-associated gene S100A4 (also known as Fibroblast-specific protein 1) after the described two-stage protocol. The significance of this difference was much greater (P < 0.001) than the increase of KLF4. Interestingly, despite the lack of evidence for the involvement of S100A4 in iPSC generation, expression of this gene has been reported in adult stem cells and cancer stem cells. Morris and collages showed that S100A4 overexpression has been associated with quiescent and undifferentiated cells in hair follicle stem cells [35]. Remarkably, in recent years, the relationship between S100A4 overexpression and cancer stem cells has become clearer. In this context, S100A4 in combination with GDF15 causes human gastric cells to adopt stem-cell-like properties [36]. Moreover, it has been observed that S100A4 can enhance the proliferation capacity of bladder cancer stem cells [37]. Furthermore, recently it has been proposed as a biomarker for glioma cancer stem cells [38]. In agreement, some intratumoral signaling crosstalk, e.g., glucose restriction, up-regulate the expression of pluripotency related genes SOX2, OCT4, and NANOG, increasing cancer stem cells [39]. Additionally, KLF4 associate directly with S100A4 to promote pulmonary metastasis by increasing myofibroblasts in the lungs [40,41]. S100A4 has a major role in tumor progress and metastatic expansion via transcription up-regulation by WNT signaling pathway [42]. This is interesting because in the cocktail reported here, the GSK3β inhibitor CHIR99021 was used. Lastly, the epithelial to mesenchymal transition (EMT) is a key event during cancer progression, and S100A4 has been reported to participate actively to mediate such transition [38,43]. In this respect, EMT and S100A4 has been linked to breast [44], lung [45], gastric [46], and uterine cancer [47]. Considering these facts, our cocktail could be activating cancer-related signaling pathways that might cause an increase in the expression of some pluripotency genes, but especially in those closely related to invasion and metastasis, such as S100A4.

Overall, we demonstrate that the combination of VPA, 5AZ, CHIR99021, and A83-01 induce the expression of some pluripotency genes, but interestingly our culture system might be stimulating the activation of genes related to cancer signaling pathways. Further experiments will be required to dissect the effect that each of the components of the formulated cocktail has and to address the specific activation of pathways. Currently, the generation of iPSC from human cells using chemical methods is not validated. It will be challenging to find an appropriate culture condition along with epigenetic modulators that could stimulate the complete endogenous reactivation of pluripotency genes, with or without the ectopic introduction and forced expression of Yamanaka factors. In this perspective, the screening of other epigenetic drugs and small molecules capable of overcoming the reprogramming barrier at early stages and also to stimulate the expression of pluripotency genes is necessary.

## 4. Materials and Methods

### 4.1. Chemicals

5-aza-2′-deoxycytidine (5AZ) (Cat. A3656), valproic acid sodium salt (VPA) (Cat. P4543), CHIR99021 (Cat. SML1046), and A83-01 (Cat. SML0788) were purchased from Sigma-Aldrich (St. Louis, MO, USA).

### 4.2. Cell Culture

Human adult dermal fibroblasts (HDF), purchased from the American Tissue Culture Collection (ATCC, PCS201012), were cultured in DMEM/F12 supplemented with 2.5 mM L-glutamine, 10% fetal bovine serum (Gibco, Carlsbad, CA, USA), 1% of non-essential amino acids, and 100 U/mL of penicillin and 100 μg/mL of streptomycin. BJ foreskin neonatal fibroblasts, purchased from the American Tissue Culture Collection (ATCC, CRL-2522), were cultured in DMEM high glucose supplemented with 2.5 mM L-glutamine, 10% fetal bovine serum, 1% of non-essential amino acids, and 100 U/mL of penicillin and 100 μg/mL of streptomycin. Cultures were maintained at 37 °C in a humified atmosphere with 5% CO_2_.

### 4.3. Reprogramming of BJ Foreskin Neonatal Human Fibroblasts

Reprogramming of BJ fibroblasts was made by transfection of episomal vectors as previously reported [15]. Briefly, 3 µg of each plasmid (pCXLE-hOct3/4-shp53, pCXLE-hSK, pCXLE-hUL, and pCXLE-eGFP) was transfected to 4 × 10^5^ BJ cells (passage 2) with a Nucleofector device (Lonza, Allendale, NJ USA) using the Amaxa Human Dermal Fibroblast Nucleofector kit (program U023) according to the supplier’s specifications. Then, cells were seeded in six-well plates pre-coated with gelatin (Sigma-Aldrich) and cultured for seven days with DMEM high glucose supplemented with 10% of fetal bovine serum, 100 U/mL of penicillin, 100 μg/mL of streptomycin, and 1 mM of VPA (Sigma-Aldrich). At day eight, cells were harvested and seeded on mitotically-inactivated mouse embryonic fibroblasts (iMEFs), and the medium was changed to KnockOut DMEM supplemented with 20% KnockOut Serum Replacement, 2.5 mM Glutamax, 0.1 mM 2-mercaptoethanol, 1% non-essential amino acids, and 10 ng/mL of basic Fibroblast growth factor (bFGF). Subsequently, the colonies were counted at 25–30 days, and those similar to human ESCs were selected for expansion and characterization. Cultures were maintained at 37 °C in a humified atmosphere with 5% CO_2_.

### 4.4. HDF Treatment With VPA, 5AZ, and Small Molecules

First, 4×10^4^ cells were seeded in six-well plates with HDF medium. After 48 h, the standard medium was removed, the cells were washed with PBS, and the Stage 1 medium was added: DMEM/F12, 2.5 mM L-glutamine, 10% fetal bovine serum, 1% non-amino acid essential, 10% Knockout Serum Replacement (Gibco), 100 U/mL of penicillin, 100 μg/mL of streptomycin, 1 mM VPA, 1 μM 5AZ, 5 μM CHIR99021, a GSK-3α inhibitor (Sigma-Aldrich), and 50 μg/mL ascorbic acid (Sigma-Aldrich) with change every 48 h. At day 10 of culture, cells were harvested and re-plated in a six-well plate. After 15 days of culture in Stage 1 medium, the medium was switched to Stage 2 medium: DMEM/F12, 2.5 mM L-glutamine, 10% fetal bovine serum, 1% non-essential amino acids, 10% Knockout Serum Replacement, 100 U/mL of penicillin, 100 μg/mL of streptomycin, 1 mM VPA, 1 μM 5AZ, 5 μM CHIR99021, 50 μg/mL ascorbic acid, 0.5 μM A83-01 (Sigma-Aldrich), and 25 ng/mL of bFGF (Invitrogen) with medium change every 48 h for additional 15 days.

### 4.5. Viability Assays

For assessment of cell viability, CellTiter-Blue Cell Viability Assay (Promega, Madison, WI, USA) was used according to the manufacturer’s instructions. HDF were seeded into 96-well plates at concentrations of 2 × 10^3^ cells per well. Every 24 h for a period of 96 h, pre-warmed Celltiter Blue reagent was added to the medium; plates were gently shaken and incubated for 3 h at 37 °C with 5% CO_2_. Before measurements, the reaction was stopped and stabilized with 3% SDS. Cell viability was estimated by fluorescence at 530_Exc_/590_Em_ nm. Data were normalized to vehicle control-treated cells (DMSO 0.1% or PBS).

### 4.6. RNA Extraction

Total RNA extraction was performed using TRIzol reagent (Ambion, Carlsbad, CA, USA) according to the manufacturer’s instructions. Briefly, the medium was removed, and cells were washed once with PBS. Then, 1 mL of TRIzol reagent was added per well of cultured cells of a six-well plate, incubated for 10 min, and transfer to a 1.5 mL microtube. Subsequently, 0.2 mL of chloroform was added and stirred for 15 s. Samples were incubated at room temperature for 2–3 min and then centrifuged at 12,000× *g* for 15 min at 4 °C. The aqueous phase was removed and placed in a new microtube. Total RNA was precipitated with 0.5 mL of absolute isopropanol and centrifuged at 12,000× *g* for 10 min at 4 °C, and an ethanol wash step was performed. Finally, samples were centrifuged at 7500× *g* for 5 min at 4 °C, and the RNA pellet was resuspended in 30 µL of nuclease-free water (Invitrogen, Carlsbad, CA, USA). The purity and concentration of total RNA were estimated using spectrophotometry at 260 and 280 nm. The RNA integrity was assessed by electrophoresis in 1% agarose gels stained with GelRed (Biotium, Hayward, CA, USA) 

### 4.7. Reverse Transcription and quantitative PCR Assays

cDNA synthesis was carried out using M-MLV reverse transcriptase (Invitrogen) and random primers (Invitrogen) according to the manufacturer’s instructions. Briefly, 1 µg of total RNA plus random primers and deoxynucleoside triphosphate were used to prepare mix 1. Mix 1 was incubated at 65 °C for 5 min and then cooled on ice. Separately, Mix 2 was prepared using First Strand buffer, DTT, and RNase OUT (Invitrogen). Mix 2 was added to Mix 1 and incubated at 37 °C for 2 min. M-MLV reverse transcriptase (200 U/µL) was then added to the reaction tube, and the following conditions were set up: incubation at 25 °C for 10 min, followed by incubation at 37 °C for 50 min, and then the enzyme was inactivated at 70 °C for 15 min. The functionality of the cDNA was confirmed by the amplification by PCR of the constitutive ribosomal 18S gene with the following primers: Forward 5′-GTT ATT TCC AGC TCC AAT AGC GTA-3′ and Reverse 5′-GAA CTA CGA CGG TAT CTG ATC GTC-3′. Quantitative PCR was performed with the 7500 Fast Real-Time PCR System (Applied Biosystem, Foster City, CA, USA) using the following gene-specific primer/probe mixes: OCT4 (POU5F1, Hs01895061_u1), SOX2 (Hs01053049_s1), NANOG (Hs02387400_g1), KLF4 (Hs00358836_m1), MYC (Hs00153408_m1), LIN28 (Hs00702808_s1), and S100A4 (Hs00243202_m1; TaqMan Gene Expression Assays, Applied Biosystems). The PCR reaction was carried out in a 20 μL volume containing 10 μL of TaqMan Universal PCR Master Mix (Applied Biosystem, Carlsbad, CA, USA), 1 μL primer/probe, 3 μL of cDNA template, and 6 μL of nuclease-free water. Amplification was performed in the standard mode using the following reaction conditions: an initial incubation at 50 °C for 2 min, followed by 95 °C for 10 min, and then 40 cycles at 95 °C for 15 s and 60 °C for 1 min. The dynamic range curve was established (from 1:16 to 1:1024 dilutions), and technical triplicates at 1:128 dilutions were included in the 96-well plates. Negative template controls were included for all assays. As an endogenous control, analysis of GAPDH (Thermo Fisher Scientific, cat 4310884E) gene expression was performed in parallel. Threshold values were used for analysis with the comparative method described by Livak and Schmittgen [48].

### 4.8. Bisulfite Sequencing

Genomic DNA from HDF at the end of the 30 days of treatment was extracted with Wizard Genomic DNA Purification kit (Promega) according to the manufacturer’s recommendations. Approximately 2 × 10^6^ cells were seeded with their respective treatment. After, cells were trypsinized, washed with PBS and lysed with 600 µL of Nuclei Lysis solution with RNAse. Protein precipitation solution was added and the sample was vortexed for 20 s at high speed. Genomic DNA was recovered by isopropanol precipitation and washed with 70% ethanol. Genomic DNA was resuspended, quantified, and visualized in agarose gel stained with GelRed (Biotium). Afterward, the bisulfite conversion of 1 µg of genomic DNA was performed using the innuCONVERT Bisulfite All-In-One Kit (Analytik, Upland, CA, USA) according to the manufacturer’s instructions. Then, 50 µL genomic DNA (1 ug), 70 µL conversion reagent, and 50 µL conversion buffer reaction were incubated at 85 °C for 45 min in a thermomixer under continuous shaking of 800 rpm. Then, the conversion reaction was spun down and the sample was transferred into a 1.5 mL reaction tube with 700 µL binding solution GS. The sample was pipetted up and down and applied to a spin filter located in a receiver tube and centrifuged at 14,000× *g* for 1 min. The spin filter was placed into a new receiver tube and 200 µL washing solution BS was added, and the sample was centrifuged at 14,000× *g* for 1 min. The spin filter was placed again in a new receiver tube and 700 µL ready-to-use desulfonation buffer was added and incubated at room temperature for 10 min. The sample was centrifuged at 14,000 rpm for 1 min. The spin filter was placed into a new receiver tube, and 500 µL washing solution C, 600 µL washing solution BS and 650 µL ethanol absolute (twice) were added sequentially among centrifugation steps at 14,000× *g* for 1 min. The spin filter was placed into an elution tube and 50 uL elution buffer was added. Samples were incubated at room temperature for 1 min and centrifuged at 8000× *g* for 1 min. Converted DNA was stored at −80 °C for further studies. Then, OCT4 promoter region was amplified by PCR using bisulfite specific primers and conditions previously reported [49]. The PCR product was subcloned into pCR2.1-TOPO-TA vector (Thermo Fisher Scientific, Boston, MA, USA). Five clones of each sample were sequenced using 3130xl Genetic Analyzer and the BigDye™ Terminator v3.1 Cycle Sequencing Kit (Applied Biosystems, Foster, CA, USA). Finally, the sequences obtained were analyzed using the software Seqscape v2.7.

### 4.9. Immunofluorescence Assays

iPSC colonies were cultured in 24-well plates with glass coverslips pre-coated with 0.5% gelatin. Subsequently, the culture medium was removed, cells were washed with PBS three times, and fixed with 4% paraformaldehyde for 10 min. Cells were permeabilized and blocked for 1 h in PBS with 0.3% Triton X-100 and 10% normal goat serum. Then, incubation with primary antibody diluted in blocking solution (PBS with 10% of normal goat serum) was done overnight at 4 °C. After washing, cells were incubated with appropriate secondary antibodies for 1 h at room temperature, counterstained with DAPI, and mounted with Aqua-Poly/Mount (Polysciences, Warrington, PA, USA). The following antibodies were used at the indicated dilutions: mouse anti-OCT4 (BD Biosciences 611202, 1:250), rabbit anti-SOX2 (Sigma-Aldrich AB5603, 1:500), and rabbit anti-NANOG (Prepotech 500-P236, 1:1000). As secondary antibodies, goat anti-mouse conjugated with Alexa Flour 568 (Thermo Fisher Scientific) and goat anti-rabbit conjugated with Alexa Flour 647 (Thermo Fisher Scientific) were used according to the supplier’s instructions at a dilution of 1:500. Immunostainings were analyzed with an epifluorescence microscope (Nikon, Eclipse TE2000-U) and photographed with a Nikon digital camera (DMX1200 F).

### 4.10. Statistical Analysis

Quantitative PCR results were analyzed with unpaired t-test with the SPSS, v. 29 software (Chicago, IL, USA). Data are shown as the means values ± standard deviation. The average of each treatment was compared, and the criterion for significance was set at *p* < 0.05 in all cases.

## 5. Conclusions

In conclusion, we present influential epigenetic properties of VPA and 5AZ in combination with CHIR99021 and A83-01 molecules that are capable of reactivating the pluripotency genes in human dermal fibroblast cells, without evident morphology changes associated to iPSC reprogramming. This preliminary approach allows to continue studying the effect of these molecules in the reprogramming efficiency of somatic cells. Additionally, we also present S100A4 increases by this cocktail, for the activation of oncogenes that might, in turn, facilitate studies of stem cell genomic networks.

## Figures and Tables

**Figure 1 molecules-26-01909-f001:**
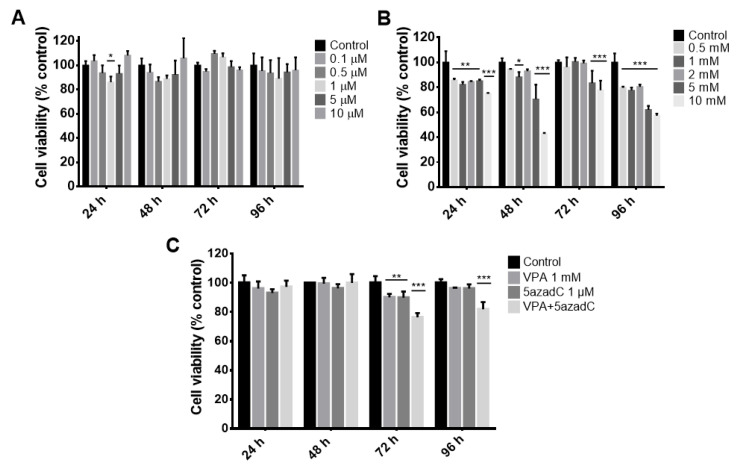
Cytotoxicity assays of epigenetic drugs 5-aza-2′-deoxycytidine and valproic acid in human dermal fibroblast cells. Dose–response–time analysis was performed to evaluate the cytotoxic effect of the epigenetic molecules 5AZ (**A**), VPA (**B**), and the combination of both (**C**). Values are expressed as mean ± SD from three independent experiments. Two-way ANOVA with Dunnett multiple comparison tests was used for comparisons between control and other groups. *, *p* < 0.05; **, *p* < 0.01; ***, *p* < 0.001.

**Figure 2 molecules-26-01909-f002:**
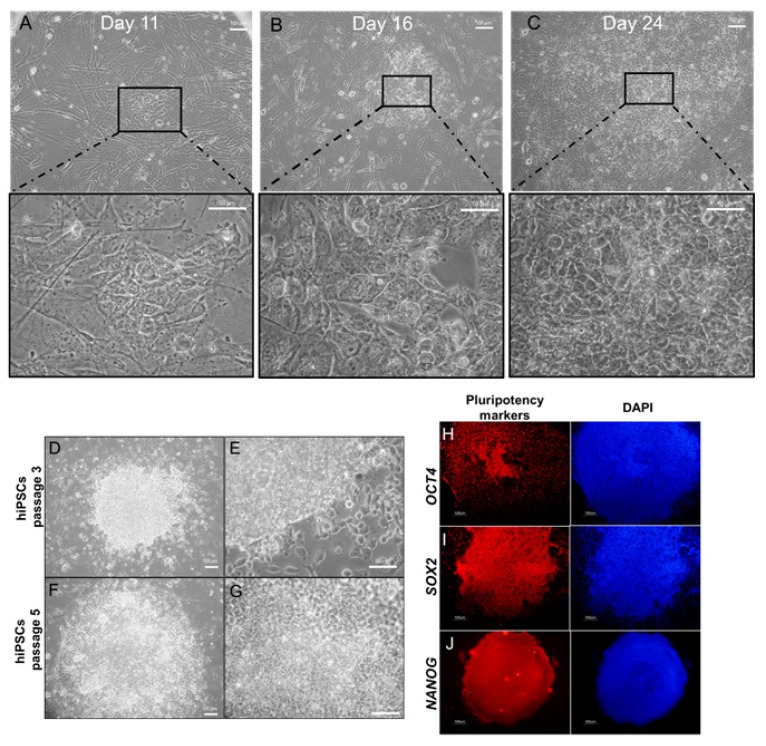
Induction of hiPSC-like colonies by episomal vectors containing the Yamanaka factors and detection of pluripotency markers in the reprogramed cells. The emergence of hiPSC-like colonies was monitored at day 11 (**A**), 16 (**B**), and 24 (**C**). Each day corresponds to a different colony. Representative images for the expansion and maintenance of hiPSC-like colonies at passage 3 (**D**) and 5 (**F**). Representative high magnification images of a hiPSC-like colony on passage 3 (**E**) and 5 (**G**). Detection of the OCT4 (**H**), SOX2 (I), and NANOG (**J**) protein by immunofluorescence on hiPSC-like colonies at passage 5 counterstained with DAPI. Scale bars represent 100 µm for A–G and 50 µm for H–J.

**Figure 3 molecules-26-01909-f003:**
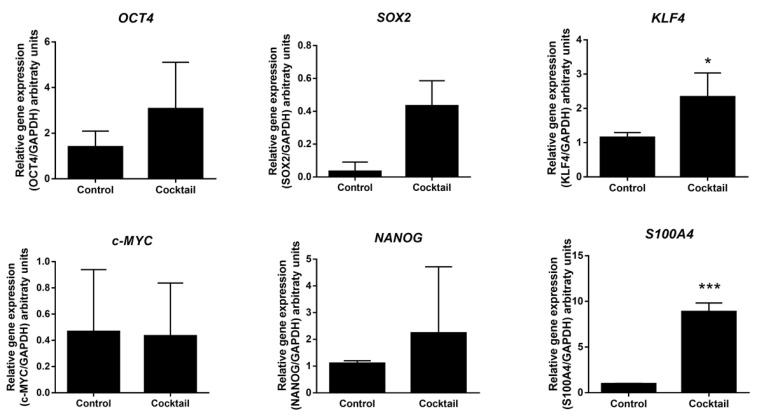
Expression of pluripotency genes after treatment of human fibroblasts with the epigenetic cocktail in the absence of the reprogramming genes. Cells were cultured for 15 days in stage 1 medium followed by 15 days in stage 2 medium, as described in Materials and Methods. Each bar represents the mean ± SD of three independent experiments, except for SOX2 gene, which was performed in duplicate. Results were normalized to GAPDH gene expression. Cocktail contains 1 mM VPA, 1 µM 5AZ, 5 µM CHIR99021, and 0.5 µM A83-01. * *p* < 0.05; *** *p* < 0.001.

**Figure 4 molecules-26-01909-f004:**
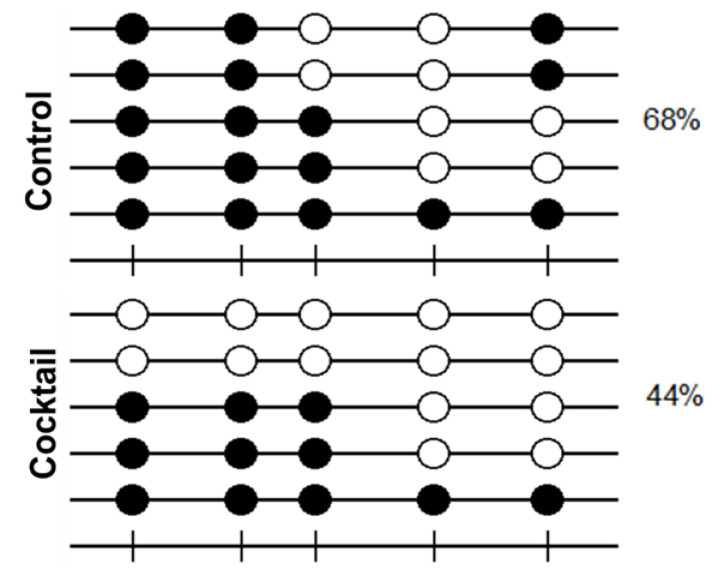
DNA methylation profile of the OCT4 promoter in human adult dermal fibroblast cells treated with 5AZ, VPA, and a cocktail of small molecules. Results were obtained from five different subcloned 221-bp PCR product for each group. Open and closed circles indicate unmethylated and methylated CpGs, respectively. Cocktail contains 1 mM VPA, 1 µM 5AZ, 5 µM CHIR99021, and 0.5 µM A83-01. Each row represents data from a single DNA molecule. The number shown in each lollipop methylation diagram indicates the percentage of methylated CpG sites.

## Data Availability

The data presented in this study are available on request from the corresponding author.

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
