# Peer review of "5-Aza-2′-Deoxycytidine and Valproic Acid in Combination with CHIR99021 and A83-01 Induce Pluripotency Genes Expression in Human Adult Somatic Cells"

_molecules, 2021, doi:10.3390/molecules26071909_

Round 1

Reviewer 1 Report

The manuscript by Aguirre-Vazquez et al. entitled with "5-aza-2′-deoxycytidine and Valproic Acid in Combination With CHIR99021 and A83-01 Induce Pluripotency Genes Expression in Human Adult Somatic Cells " reported the effect of two epigenetic enhancer compounds, 5AZ and VPA.

Although the cocktail of the compounds that the authors examined here sounds new and had a certain effect for increasing gene expression of Yamanaka factors, this reviewer found serious concerns over the construction of the experiments and I believe some of the authors' representation of the results to be flawed.

Major issues

  1. The authors’ idea that the use of 5AZ in combination of VPA seems reasonable and rational, since the former is a DNA methyltransferase inhibitor, and the latter is HDAC inhibitor. Both the enzymes are critical for controlling epigenetic status of the target cell genome. However, the combination of these class of inhibitors itself is not new and very popular in the related field. For example, there are many reports that use trichostatin A (TSA) as a HDAC inhibitor in combination with 5AZ. Thus, the authors may better show the merit of use of VPA instead of TSA. This point may enhance the novelty and the technical merit.

  1. Missing several important negative controls.

The reviewer is afraid that the statement in the abstract of this manuscript is not completely supported by the authors’ experiments in the study. In the figure 3, the expression levels of Yamanaka factors and the other two genes were shown, only the conditions with and without the cocktail. However, CHIR99021 and A83-01 have strong signaling modulator, in which the former is the canonical Wnt signaling and the later is the TGF signaling. Thus, at least the cells treated with these two inhibitors without the epigenetic modulators is needed as an additional negative control. Since the result is partly negative, which failed to induce pluripotency of human somatic cells, the analysis of the negative result is especially important to discuss why the result is negative.

  1. Accordingly, since both the Wnt agonist CHIR and TGF inhibitor A83-01 can repress several DNA methyltransferases and HDACs, additional negative control should be added for the experiments Figure 4. Bisulfite sequencing of OCT4 gene of the cells treated only with CHIR and A83-01, and the cocktail without VPA as the two additional negative control are quite important, otherwise the effect of VPA cannot be discussed.

  1. Missing detailed experimental protocol and the data of bisulfite sequencing. The authors need to prepare the method section and result section in detail. In the Figure 4, there are the five circle positions with open and closed circles. The author describe which position is corresponding to the CpG site of the OCT4 promoter sequences. There is no detailed information in the reference 46 for the readers who likes to reproduce and evaluate the results of this manuscript.

  1. A further analyses of bisulfite sequencing for the other genes, especially for NANOG and S100A4, seems very important to conclude the authors claims in the abstract. Thus, addition of these data is strongly recommended.

Minor comments

  1. There are several important key references are not properly referred. At least the following two reference, the combination of 5AZ with TSA and the importance of the epigenetic control of the OCT4 gene expression should be included.

Increased pre-implantation development of cloned bovine embryos treated with 5-aza-2'-deoxycytidine and trichostatin A.

Ding X, Wang Y, Zhang D, Wang Y, Guo Z, Zhang Y.

Theriogenology. 2008 Sep 1;70(4):622-3

Epigenetic control of mouse Oct-4 gene expression in embryonic stem cells and trophoblast stem cells.

Hattori N, Nishino K, Ko YG, Hattori N, Ohgane J, Tanaka S, Shiota K.

J Biol Chem. 2004 Apr 23;279(17):17063-9

Reviewer 2 Report

Dear editor and authors,

I have several comments and questions.

Line 92. How tested concentrations of reprogramming molecules were selected?

Line 129. What are "gradual changes that occur during the reprogramming process"? What are specific cell characteristics to predict that they are going to become pluripotent? To my knowledge, there is no way to predict whether a particular cell during culture would become iPS cell based on morphology. And such reliable and measurable changes are not described above. For instance, "the formation of epithelial-like cell aggregates was observed". How many aggregates? How many become iPS cell colonies? Is epithelial-like cell morphology a valid predictor of future reprogramming success?

Line 143. The fibroblasts treated with small molecules were not losing fibroblast morphology and there were no colonies. That raises a question: Why the experiment of iPSC generation and their morphology are described? The part with iPSС generation does not provide any new information and, in my opinion, is not relevant to the study and should be removed.

Line 160. What is the interpretation of hypervariable expression of c-MYC and NANOG? To my knowledge, c-MYC is expressed in both fibroblasts and iPSCs, it is not a unique "pluripotency" gene. And there is no difference in its expression (Fig. 3). As for NANOG, to my knowledge it is not expressed in fibroblasts. Of course its expression may be "leaking", but the 2-fold difference in expression levels (Fig. 3) is probably just technical error, as the lower expression level is, the bigger is the error.

Line 167. "Results were normalized to GAPDH gene expression". We see very slight differences in expression levels of OCT4 and KLF4. It is necessary to use several genes for normalization to exclude expression variability of these genes.

Line 175. To reveal CpG-methylation levels more than 30 individual molecules should be analyzed. Contemporary sequencing methods allow simultaneous analysis of thousands of molecules. Data on 5 molecules is absolutely insufficient.

Line 186. I do not agree with the conclusion of significantly increased the expression": SOX2 expression was measured in duplicate, thus statistical analysis was not performed, the difference of KLF4 expression is 2-fold. That is close to the qPCR limit about 1.5 folds. ddPCR would give more reliable data. As for OCT4 and NANOG - I'd explain the differences by technical error.

In conclusion, not enough reliable new data is presented in the article.

Reviewer 3 Report

Comments:

  1. On Figure 1, the authors showed the toxicity assays up to 96 hours. Based on the procedure on Figure 2, the toxicity assay needs to be expanded to 15-24 days.
  2. On Figure 1, it seems that 5AZ has some protective effect when used in combination of 5AE and VPA. Please explain. This is why I also suggest toxicity assay up to 15 to 24 days.
  3. On Figure 3, please show western blot analysis on all proteins tested.
  4. On Figure 3, it is not clear if the experiments were done at 48 hours after treatment.
  5. On Figure 4, please indicate how many times of experiments were performed. Also is 68% vs 44% significant in statistical analysis?

Round 2

Reviewer 2 Report

Dear editor and authors,

Some of my comments were answered and clarified.

I still do not agree with some points presented in the MS:

Line 116 – Section “2.2. Morphological Changes by Epigenetic Molecules”

The authors produced iPSCs from human dermal fibroblast and explained in the text and reply to reviewer that it was a control. They were able to observe morphological changes to cell morphology due to mesenchymal-to-epithelial transition. That information is not new, and in my opinion, a reference to the article with the morphological study of this transition would be enough. The fibroblasts after the cocktail usage had not changed the morphology, thus the experimental proof of such changes in iPSC generation is not necessary.

Lines 158 and 175 – probably “2.3” and “2.4” instead of “4.3” and “4.4”.

Line 164. The interpretation of the differences between gene expression is, in my opinion, incorrect. “A hypervariable expression” of c-MYC could be the case, as c-MYC known to be expressed in fibroblasts. As for NANOG, human fibroblasts are not supposed to express NANOG. When extremely low (leaking?) expression is compared with similarly low expression after the cocktail exposure, “drastical” changes are expected due to random fluctuation of expression. “Fluctuation” of gene expression is expected when it is on “noise” level. It is meaningless from biological point of view.

Line 186.

“Comment: Line 175. To reveal CpG-methylation levels more than 30
individual molecules should be analyzed. Contemporary sequencing methods
allow simultaneous analysis of thousands of molecules. Data on 5
molecules is absolutely insufficient.
Response: Regarding the number of replicates for this type of
assay, we used as references to Li et al., 2009 and Zhu et al.,
2010 papers, where 4-6 replicates were included in the methylation
profiles for OCT4 promoter.
1. Li W, Wei W, Zhu S, et al.: Generation of rat and human induced
pluripotent stem cells by combining genetic reprogramming and
chemical inhibitors. Cell Stem Cell 4: 16-19, 2009.
2. Zhu S, Li W, Zhou H, et al.: Reprogramming of human primary
somatic cells by OCT4 and chemical compounds. Cell Stem Cell 7:
651-655, 2010.”

I completely disagree with the response “other researchers used the same number of replicates”. I can find articles with tenths of thousands strands sequenced. The problem here is biological – the CpG methylation is often stochastic, so when you analyze just 5 samples you would have a huge bias. The authors analyzed just 25 CpG islands for control and cocktail. The raw data is 8:17 CpG-:CpG+ for control and 14:11 for cocktail. The real difference is not 26% as claimed in the Figure. As the methylation is stochastic, if any strand would be substituted with another from the treated sample, the percentage would shift drastically. Much higher number of samples (strands in this case) must be used to minimize the statistical bias. 5 is absolutely not enough.

Line 193.

“Comment: Line 186. I do not agree with the conclusion of significantly
increased the expression": SOX2 expression was measured in duplicate,
thus statistical analysis was not performed, the difference of KLF4
expression is 2-fold. That is close to the qPCR limit about 1.5 folds.
ddPCR would give more reliable data. As for OCT4 and NANOG - I'd explain
the differences by technical error.
Response: As we have commented previously, the expression analyzes
have been carried out based on the standards described for
quantitative assays.”

I do not agree that “the expression analyzes have been carried out based on the standards described for
quantitative assays.” In case of SOX2, it is stated in the text, that the analysis was performed in duplicate, not in triplicate. That is probably the reason, that the statistical significance was not revealed. Thus, the data on SOX2 expression are not reliable.

Line2 217-219. “These findings indicate that the 5AZ, which was included in the formulated cocktail, promotes the demethylation of OCT4 promoter region and prevents the complete silencing of OCT4 in somatic cells.”

5AZ causes demethylation, that’s a previously reported fact. In this research methylation analysis was not performed correctly, as I explained above. So the conclusion on OCT4 promoter demethylation is not based on the presented data and, thus, should not be included into the Discussion section.

There are some grammatical mistakes in the newly added parts of the MS that should be corrected.

In conclusion, not enough reliable new data is presented in the article.

Reviewer 3 Report

No more comments